# PCV2 Infection Upregulates SOCS3 Expression to Facilitate Viral Replication in PK-15 Cells

**DOI:** 10.3390/v17081081

**Published:** 2025-08-05

**Authors:** Yiting Li, Hongmei Liu, Yi Wu, Xiaomei Zhang, Juan Geng, Xin Wu, Wengui Li, Zhenxing Zhang, Jianling Song, Yifang Zhang, Jun Chai

**Affiliations:** 1College of Veterinary Medicine, Yunnan Agricultural University, Kunming 650051, China; liyiting202506@163.com (Y.L.); liuhongmei.ll@hotmail.com (H.L.); zwuyi2001cn@163.com (Y.W.); xm2224017082@163.com (X.Z.); 18687490587@163.com (J.G.); wuxinddl@163.com (X.W.); wenguili@yeah.net (W.L.); 2Yunnan Tropical and Subtropical Animal Virus Disease Laboratory, Yunnan Animal Science and Veterinary Institute, Kunming 650224, China; zhenxing978@163.com (Z.Z.); song@hotmail.com (J.S.)

**Keywords:** porcine circovirus type 2, suppressor of cytokine signaling 3, cytokine regulation, porcine kidney 15 cell line, viral replication

## Abstract

Porcine circovirus type 2 (PCV2) is a globally prevalent swine pathogen that induces immunosuppression, predisposing pigs to subclinical infections. In intensive farming systems, PCV2 persistently impairs growth performance and vaccine efficacy, leading to substantial economic losses in the swine industry. Emerging evidence suggests that certain viruses exploit Suppressor of Cytokine Signaling 3 (SOCS3), a key immune checkpoint protein, to subvert host innate immunity by suppressing cytokine signaling. While SOCS3 has been implicated in various viral infections, its regulatory role in PCV2 replication remains undefined. This study aims to elucidate the mechanisms underlying the interplay between SOCS3 and PCV2 during viral pathogenesis. Porcine SOCS3 was amplified using RT-PCR and stably overexpressed in PK-15 cells through lentiviral delivery. Bioinformatics analysis facilitated the design of three siRNA candidates targeting SOCS3. We systematically investigated the effects of SOCS3 overexpression and knockdown on PCV2 replication kinetics and host antiviral responses by quantifying the viral DNA load and the mRNA levels of cytokines. PCV2 infection upregulated SOCS3 expression at both transcriptional and translational levels in PK-15 cells. Functional studies revealed that SOCS3 overexpression markedly enhanced viral replication, whereas its knockdown suppressed viral proliferation. Intriguingly, SOCS3-mediated immune modulation exhibited a divergent regulation of antiviral cytokines: PCV2-infected SOCS3-overexpressing cells showed elevated IFN-β but suppressed TNF-α expressions, whereas SOCS3 silencing conversely downregulated IFN-β while amplifying TNF-α responses. This study unveils a dual role of SOCS3 during subclinical porcine circovirus type 2 (PCV2) infection: it functions as a host-derived pro-viral factor that facilitates viral replication while simultaneously reshaping the cytokine milieu to suppress overt inflammatory responses. These findings provide novel insights into the mechanisms underlying PCV2 immune evasion and persistence and establish a theoretical framework for the development of host-targeted control strategies. Although our results identify SOCS3 as a key host determinant of PCV2 persistence, the precise molecular pathways involved require rigorous experimental validation.

## 1. Introduction

Since its initial isolation in 1998, porcine circovirus type 2 (PCV2) has emerged as a significant global threat to swine production, resulting in substantial economic losses [1]. PCV2 is a non-enveloped, icosahedral, single-stranded, circular DNA virus characterized by a remarkably compact genome of approximately 1.7 kb. The virus replicates by hijacking host S-phase proteins, leading to the formation of pathognomonic cytoplasmic or intranuclear inclusion [2]. Among its 11 open reading frames (ORFs), the proteins encoded by ORF1 (Rep) and ORF2 (Cap) are crucial for viral replication and capsid assembly, respectively, and also function as key immunogenic targets [3,4,5,6]. PCV2 remains one of the most economically significant swine pathogens worldwide. Although the PCV2d-based vaccines licensed in 2024 have reduced clinical disease and improved production parameters, their long-term effectiveness is increasingly threatened by the virus’s exceptional evolutionary rate. PCV2 encompasses eight recognized genotypes (PCV2a–PCV2h), with PCV2a, PCV2b, and the emergent PCV2d predominating in global swine populations; PCV2c, PCV2e, PCV2f, PCV2g, and PCV2h circulate at markedly lower frequencies. Notably, PCV2 exhibits the highest recorded nucleotide substitution rate (1.2 × 10^−3^ substitutions/site/year) among all single-stranded DNA viruses, facilitating a rapid antigenic drift. Historical epidemiological data reveal two major genotype shifts preceding widespread vaccine use [7]. Prior to the introduction of the first commercial PCV2 vaccines in the mid-2000s, the global dominant strain transitioned from PCV2a to PCV2b, with the latter demonstrating enhanced virulence and tissue tropism. More recently, a second transition has occurred, with PCV2d supplanting PCV2b as the predominant genotype in North America, Asia, and Europe. Alarmingly, PCV2d isolates are now routinely recovered from herds with documented vaccination compliance, raising the possibility that this genotype has acquired immune-escape mutations capable of circumventing vaccine-induced protection. Understanding the genetic and antigenic basis of this emerging genotype is therefore critical for the rational design of next-generation PCV2 vaccines [1]. PCV2-associated disease (PCVAD) includes both overt clinical syndromes, such as the postweaning multisystemic wasting syndrome (PMWS) and porcine respiratory disease complex (PRDC), as well as subclinical infections [8]. Importantly, subclinical PCV2 infections, despite the absence of overt pathology, significantly compromise growth performance, feed efficiency, and vaccine responsiveness, resulting in substantial economic burdens [9]. Crucially, PCV2 subverts host immunity through cytokine dysregulation, which predisposes the host to secondary infections and viral persistence. This immunosuppressive strategy is further evidenced by its tropism for immune cells (e.g., lymphocytes, alveolar macrophages) and lymphoid tissues, highlighting unresolved questions regarding its immune evasion mechanisms [10,11,12].

The suppressor of cytokine signaling (SOCS) family comprises critical feedback inhibitors that fine-tune immune responses by attenuating cytokine signaling cascades [13,14,15,16]. Among these, SOCS3 acts as a master regulator of the JAK-STAT pathway, employing its SH2 domain to bind to phosphorylated receptors and its kinase inhibitory region (KIR) to suppress JAK activity, thereby serving as a molecular brake on inflammatory responses [17,18]. Mounting evidence indicates that viral pathogens, including HIV and HCV, exploit SOCS3 to evade host antiviral defenses and establish persistent infections [19,20,21,22].

The present study elucidates the regulatory role of SOCS3 in PCV2 infection. A PK-15 cell line stably over-expressing porcine SOCS3 was generated via lentiviral transduction, whereas transient SOCS3 silencing was achieved by siRNA transfection. These complementary gain- and loss-of-function models were employed to systematically evaluate the impact of SOCS3 abundance on PCV2 replication kinetics. Concomitantly, the transcriptional profiles of key antiviral IFN-β and TNF-α were quantified by real-time RT-PCR to delineate the cytokine milieu modulated by SOCS3 during PCV2 infection. Collectively, our data not only clarify the functional contributions of SOCS3 to PCV2 replication in PK-15 cells but also provide mechanistic insights into the SOCS3-mediated regulation of innate immune responses, thereby offering a theoretical framework for understanding the molecular basis of PCV2 subclinical persistence.

## 2. Materials and Methods

### 2.1. Cells Virus and Reagents

Porcine kidney cells (PK-15) and human embryonic kidney epithelial cells (HEK-293T) were maintained by the Microbiology Laboratory of Yunnan Agricultural University (Kunming, China). The PCV2d strain KM-2018 (GenBank accession number: MK405698), isolated from the Yunnan province, was also preserved in the same institution. These cells were cultured in Dulbecco modified Eagle medium (DMEM, Gibco, Thermo Fisher Scientific, Waltham, MA, USA). All media were supplemented with 10% heat-inactivated fetal bovine serum (FBS, biological industries, Beit Haemek, IL, USA) and maintained at 37 °C with 5% CO_2_. The PCV2 strain (GenBank accession no. MK405698), which belongs to the PCV2d gene subtype, was isolated and stocked in our lab.

The chloroform solution, isopropyl alcohol solution, anhydrous ethanol solution, and other reagents were purchased from Chongqing Chuandong Chemical Industry Group (Chuandong, Chongqing, China). A viral DNA extraction kit and 2 × ApexHF FS PCR Master Mix (dye plus) were purchased from Accurate Biotechnology (Changsha, China) Co., Ltd. An All-In-One 5X RT Master Mix (with an AccuRTGenomic DNA Removal Kit) was purchased from Applied Biological Materials Inc. (ABM, Richmond, BC, Canada). The pMD19-T Vector (pMD19-T), Escherichia coli strain DH5α (DH5α), restriction endonuclease Xba I and Not I, T4 DNA ligase, and RNAiso Plus were purchased from TaKaRa Biomedical Technology (Beijing) Co., Ltd. (Takara, Shiga, Japan).

### 2.2. Plasmid Constructions

The encoding gene of porcine SOCS3 (GenBank accession number 493186) was amplified from the cDNA of PK-15 cells, subcloned into the pMD19-T (Takara Biomedical Technology, Beijing, China), and transformed into *E. coli* competent cells. The positive clones were selected via blue–white screening and further verified by colony PCR. Sanger sequencing confirmed the correct insertion of the target fragment, and the validated recombinant plasmid was designated as pMD19-T-SOCS3. The PCR primers used in this study are shown in Table 1. All sequences in the constructs were confirmed by sequencing analysis (Sangon Biotech, Shanghai, China).

### 2.3. siRNA and Transfection

Specific siRNA target sites against SOCS3 mRNA were designed, and three siRNA sequences (designated as SOCS3-si-1, -2, and -3), along with a scrambled negative control siRNA (si-NC), were synthesized (Table 2). For transfection optimization, PK-15 cells (4 × 10^5^ cells/well) were seeded in 12-well plates. The transfection complexes were prepared using LipoRNAi™ Transfection Reagent (Beyotime Biotechnology, Shanghai, China) and Cy3-labeled siRNA at the gradient concentrations (0–100 nM). The transfection efficiency was evaluated via fluorescence microscopy imaging at 48–72 h post-transfection to determine the optimal siRNA concentration. For validation of the silencing efficacy, PK-15 cells (2 × 10^5^ cells/well) in 24-well plates were transfected with the optimized siRNA concentration. The dynamic changes in the SOCS3 mRNA expression levels were monitored at 12–72 h post-transfection using qRT-PCR. All experiments included three independent biological replicates, and the data were expressed as the mean.

### 2.4. Construction of the SOCS3-Overexpressing PK-15 Cell Line

The recombinant plasmid pCDH-CMV-SOCS3 was constructed via molecular cloning. Porcine SOCS3 cDNA was ligated into the linearized pCDH-CMV-MCS-EF1-GFP + Puro vector (System Biosciences, Palo Alto, CA, USA) using T4 DNA ligase (Takara Biomedical Technology, Beijing, China). The ligation product was transformed into Stbl3 competent cells (TransGen Biotech, Beijing, China). The positive clones were selected by ampicillin resistance screening and validated through colony PCR and double restriction enzyme digestion (EcoRI/XbaI, Thermo Fisher Scientific, Waltham, MA, USA). Plasmid sequencing (Sangon Biotech, Shanghai, China) confirmed the sequence integrity.

Endotoxin-free plasmids were co-transfected with lentiviral packaging plasmids pMD2.G and psPAX2 (Addgene, Watertown, MA, USA) into HEK-293T cells using the Lipo8000™ transfection reagent (Beyotime Biotechnology, China). The lentiviral particles were harvested at 48–72 h post-transfection and concentrated by ultracentrifugation. PK-15 cells were infected with viral supernatants and selected with 5 μg/mL puromycin (Solarbio, Beijing, China) for 7 days to establish stable cell lines: SOCS3-overexpressing (PK-15-SOCS3) and empty vector control (PK-15-pCDH).

### 2.5. Cell Counting Kit-8 Assay

PK-15 cells in good growth condition were seeded in 96-well plates at a density of 3 × 10^4^ cells/well. For the siRNA transfection experiments, the cells were transfected with si-SOCS3-2 or negative control si-NC (RiboBio, Guangzhou, China) when reaching 30–50% confluency. For SOCS3-overexpressing cell lines (PK-15-SOCS3, PK-15-pCDH, and wild-type PK-15), the cells were directly plated at the same density.

Each experimental group, including blank controls (complete medium only), was set up in nine replicates. The cells were cultured at 37 °C with 5% CO_2_ for 24 h, 48 h, and 72 h. Three hours before the endpoint, 10 µL of CCK-8 solution (Dojindo Laboratories, Kumamoto, Japan) was added to each well, followed by 3 h of incubation at 37 °C in the dark. After gentle shaking (5 min), the absorbance at 450 nm (OD450) was measured using a microplate reader (Epoch 2, BioTek, Winooski, VT, USA). The cell growth curve was plotted based on the OD values after blank subtraction. Statistical analysis was performed using GraphPad Prism 9.0 (USA), with the data presented as the mean ± SD from three independent experiments.

### 2.6. Viral Infection Assay

PK-15-SOCS3 (SOCS3-overexpressing), PK-15-pCDH (empty vector control), and parental PK-15 cells were seeded in 24-well plates at a density of 2 × 10^5^ cells/well (triplicate wells per group) and cultured under standard conditions (37 °C, 5% CO_2_) until reaching 70–80% confluency. The cells were washed twice with PBS before infection with porcine circovirus type 2 (PCV2) at a multiplicity of infection (MOI) of 1. After 1.5 h of viral adsorption, the inoculum was replaced with maintenance medium.

In parallel experiments, PK-15 cells at 30–50% confluency were transfected with either si-SOCS3-2 or negative control siRNA (si-NC) using Lipofectamine 3000 (Thermo Fisher Scientific). At 48 h post-transfection, the cells were similarly infected with PCV2 as described above. At 24 h, 48 h, and 72 h post-infection (hpi), the cells were harvested and subjected to three freeze–thaw cycles for complete lysis. Both the cell lysates and culture supernatants were collected and stored at −80 °C for subsequent analysis. The entire experiment was performed in three independent biological replicates, with appropriate uninfected controls included in each replicate.

### 2.7. Quantitative Real-Time PCR Analysis

Total RNA was extracted from cells using the RNAiso Plus reagent (Takara Bio, Beijing, China) following the manufacturer’s protocol. cDNA synthesis was performed with 1 μg of total RNA using the Evo M-MLV RT-PCR Kit (Accurate Biotechnology, Changsha, China) while PCV2 genomic DNA was isolated using the SteadyPure Viral DNA/RNA Extraction Kit (Accurate Biotechnology). Quantitative PCR was conducted using the SYBR Green Pro Taq HS Premix III (Accurate Biotechnology) on a StepOnePlus Real-Time PCR System (Thermo Fisher Scientific, Waltham, MA, USA) under the following conditions: 95 °C for 30 s, followed by 40 cycles of 95 °C for 5 s and 60 °C for 30 s, with a melting curve analysis to confirm amplification specificity.

For the SOCS3 expression analysis, the relative mRNA levels in PCV2-infected or mock-infected PK-15 cells were normalized to β-actin and calculated using the 2^^(-ΔΔCt)^ method, with comparisons made between the PK-15-SOCS3 stable cells, empty vector controls (PK-15-pCDH), and parental PK-15 cells. In the siRNA knockdown experiments, PK-15 cells (2 × 10^5^ cells/well) were transfected with SOCS3-targeting siRNAs (si-SOCS3-1/-2/-3) or negative control siRNA (si-NC) using the LipoRNAi™ Transfection Reagent (Beyotime Biotechnology), and the knockdown efficiency was calculated as [(si-NC expression-treatment group expression)/si-NC expression] × 100%, with monitoring at 12–72 h post-transfection. The PCV2 viral load was absolutely quantified using virus-specific primers and a standard curve (10^2^–10^8^ copies/μL).

All experiments included three biological replicates with technical triplicates and no-template controls. The primer sequences are provided in Table 1. Statistical analysis was performed using GraphPad Prism 9.0, with the data presented as the mean ± SD and significance determined by one-way ANOVA with Tukey’s post hoc test (*p* < 0.05).

### 2.8. Western Blot Analysis

Cellular proteins were extracted using RIPA (radio-immunoprecipitation assay) lysis buffer supplemented with protease and phosphatase inhibitors. Following centrifugation at 12,000× *g* for 15 min at 4 °C, the supernatant was collected for protein quantification using the BCA assay. Briefly, the samples and standards were incubated with the BCA working reagent (200 μL) at 37 °C for 30 min, and the absorbance at 562 nm was measured using a microplate reader to generate a standard curve.

Protein samples were mixed with 5 × reducing loading buffer (4:1 ratio) and denatured by boiling. Electrophoresis was performed using polyacrylamide gels with an initial voltage of 80 V, which was increased to 150 V after the bromophenol blue dye migrated out of the stacking gel. The proteins were then transferred to polyvinylidene difluoride (PVDF) membranes using the wet transfer method (110 mA, 90 min). M\The membranes were blocked with 5% BSA for 2 h at room temperature.

Immunoblotting was performed using a rabbit polyclonal anti-SOCS3 antibody (Proteintech; 1:1000 dilution) incubated overnight at 4 °C, followed by incubation with a HRP-conjugated goat anti-rabbit secondary antibody (Servicebio; 1:3000 dilution) for 1 h at room temperature. The membranes were washed three times (10 min each) with Tris-buffered saline with 0.1% Tween-20 (TBST) between antibody incubations. Protein bands were visualized using an ECL chemiluminescence system, and band intensities were quantified using the ImageJ software v1.8.0. to determine the relative protein expression levels.

### 2.9. Indirect Immunofluorescence Assay (IFA)

PK-15 cells and stable transfectants (PK-15-SOCS3 and PK-15-pCDH) were seeded in 6-well plates at 1 × 10^6^ cells/well and cultured under standard conditions (37 °C, 5% CO_2_). At 30–50% confluency, the cells were transfected with either si-SOCS3-2 or negative control siRNA (si-NC) using Lipofectamine 3000 (Thermo Fisher Scientific), followed by 48 h of incubation. When the cells reached 70–80% confluency, they were infected with PCV2 (MOI = 1) for 1.5 h, with uninfected cells serving as negative controls. After replacing the inoculum with maintenance medium, the cells were cultured for an additional 36 h.

For immunostaining, cells were fixed with 4% paraformaldehyde for 20 min at room temperature and permeabilized with 0.2% Triton X-100 (Takara Bio, China) for 15 min. Non-specific binding was blocked with 1% BSA for 1 h at room temperature. The samples were then incubated with a primary antibody (rabbit anti-PCV2 replicase antibody, GeneTex; 1:1000 dilution) overnight at 4 °C, followed by a CoraLite 594-conjugated goat anti-rabbit IgG(H+L) secondary antibody (Proteintech; 1:500 dilution) for 1 h at 37 °C in the dark. After three washes with PBST, the nuclei were counterstained with DAPI (Beyotime Biotechnology) and slides were mounted for imaging.

Fluorescence images were acquired using a confocal microscope (Leica TCS SP8) with consistent exposure settings across all samples. The experiment included three independent biological replicates, with at least five random fields captured per condition. An image analysis was performed using the ImageJ software (NIH) v1.8.0. to quantify the fluorescence intensity.

## 3. Results

### 3.1. PCV2 Infection Upregulates SOCS3 Expression in PK-15 Cells

To investigate whether PCV2 infection affects SOCS3 expression, we analyzed SOCS3 mRNA and protein levels in PCV2-infected PK-15 cells. RT-PCR analysis revealed a significant upregulation of SOCS3 mRNA during the early infection stages (Figure 1a). Compared with the mock-infected controls, the SOCS3 transcript levels showed marked elevations at 24 and 48 h post-infection (hpi) (Figure 1a).

A Western blot analysis demonstrated corresponding changes in SOCS3 protein expression (Figure 1b). PCV2 infection induced progressive SOCS3 protein accumulation, peaking at 24 hpi before declining at 48 hpi (Figure 1b). These findings collectively demonstrate that PCV2 infection upregulates both SOCS3 mRNA and protein expression in vitro.

### 3.2. SOCS3 Expression Facilitates PCV2 Replication

This study systematically investigated the regulatory role of SOCS3 in PCV2 infection through genetic intervention and viral replication dynamics (Figure 2). A SOCS3 knockdown model was first established using siRNA-mediated gene silencing. Three specific siRNA sequences targeting porcine SOCS3 were designed and screened based on bioinformatic analysis, alongside a negative control. Multidimensional validation via fluorescence microscopy, qRT-PCR, and Western blot identified 50 nM as the optimal siRNA transfection concentration (Figure 2a). Among the tested sequences, si-SOCS3-2 significantly reduced SOCS3 mRNA levels (Figure 2b,c) and protein expression (Figure 2h) at 48 h post-transfection, with CCK-8 assays confirming no cytotoxicity (Figure 2g).

To complement this model, a Flag-SOCS3 stable overexpression cell line (PK-15-SOCS3) was constructed using a lentiviral vector system (Figure 2d,e). The qRT-PCR analysis demonstrated a 70-fold increase in the SOCS3 mRNA compared with the wild-type PK-15 cells (Figure 2f), with the Western blot confirming substantial protein upregulation (*** *p* < 0.001, Figure 2i) and unaltered cell viability (Figure 2g).

A viral replication analysis revealed that SOCS3 expression critically regulated PCV2 propagation. Immunofluorescence showed a marked reduction in PCV2-positive cells in SOCS3-silenced groups and enhanced signals in overexpression groups at 36 h post-infection (hpi; Figure 2k). Viral kinetics demonstrated significant suppression in SOCS3-silenced cells, with PCV2 copy numbers sharply decreasing at 48 and 72 hpi (*** *p* < 0.001) but remaining unchanged at 24 hpi (Figure 2i). Conversely, SOCS3 overexpression potentiated viral replication, with significantly elevated copy numbers throughout the infection cycle (24/48/72 hpi; *** *p* < 0.001, Figure 2j), consistent with the time-dependent viral accumulation observed in the temporal qPCR assays.

In conclusion, this study establishes SOCS3 as a key enhancer of PCV2 replication in PK-15 cells. The positive correlation between SOCS3 expression and viral replication efficiency—silencing inhibits viral proliferation while overexpression amplifies it—provides novel mechanistic insights into PCV2 pathogenesis, advancing our understanding of the host–virus interactions in this system.

### 3.3. SOCS3 Modulates PCV2-Induced Expression of IFN-β and TNF-α mRNA

SOCS3 is known to suppress antiviral responses through the negative feedback regulation of interferon signaling. To investigate whether this mechanism operates during PCV2 infection and elucidate SOCS3’s role in modulating host inflammatory responses, we dynamically monitored the transcriptional changes in the type I interferon (IFN-β) and proinflammatory cytokine (TNF-α) in SOCS3-modified PK-15 cells using RT-qPCR (Figure 3). The experimental groups included PCV2-infected PK-15 cells with either SOCS3 overexpression (PK-15-SOCS3) or knockdown (via si-SOCS3-2), along with appropriate controls (PK-15-pCDH and si-NC). Cells were harvested at 6, 12, 24, and 48 h post-infection (hpi) for mRNA quantification.

Notably, the SOCS3 knockdown resulted in significantly reduced IFN-β mRNA expression (*** *p* < 0.001) but markedly elevated TNF-α levels (*** *p* < 0.001) at all time points examined (Figure 3a,b). Conversely, the PK-15-SOCS3 cells showed a substantial upregulation of IFN-β transcripts (*** *p* < 0.001) at 6, 12, and 48 hpi (Figure 3c), while TNF-α expression was dramatically suppressed throughout the infection course (*** *p* < 0.001) (Figure 3d). The 24 hpi time point showed no significant difference in IFN-β expression in the overexpressing cells.

These findings demonstrate that SOCS3 exerts bidirectional immunomodulatory effects during PCV2 infection: its overexpression potentiates IFN-β-mediated antiviral signaling while suppressing TNF-α-driven inflammation, whereas SOCS3 deficiency weakens antiviral responses and exacerbates inflammatory activation. This dual regulatory capacity positions SOCS3 as a critical balancer between viral replication and host immune homeostasis during PCV2 infection.

## 4. Discussion

Porcine circovirus type 2 (PCV2) manipulates host immune regulatory mechanisms to establish a persistent and subclinical infection [23]. Upon viral invasion, the host cells trigger antiviral responses like innate immunity, autophagy, and the ubiquitin–proteasome system [24,25]. Nevertheless, viruses exploit host factors to aid infection, with certain cellular proteins pivotal in this process. The Suppressor of Cytokine Signaling (SOCS) protein acts as an intrinsic negative regulator of innate and adaptive immunity [26]. It is upregulated in response to viral infections, dampening the host immune reaction. SOCS3, specifically, is upregulated post-infection by the Newcastle disease virus, duck hepatitis virus type 1, and porcine reproductive and respiratory syndrome virus (PRRSV), being exploited by these viruses to evade host immune surveillance and boost viral replication [27,28,29]. This investigation revealed that PCV2 infection markedly increased SOCS3 expression in PK-15 cells, with SOCS3 levels correlating closely with virus replication efficiency: augmenting SOCS3 significantly raised PCV2’s viral load, while silencing SOCS3 via siRNA effectively curtailed virus proliferation. These findings underscore the pivotal role of SOCS3 in fostering virus replication during PCV2 infection. SOCS3 regulates host immune responses through a dual pathway: First, as a classical negative regulator of the JAK-STAT signaling pathway, SOCS3 can interfere with type I interferon (IFN-α/β)-mediated antiviral signaling [30,31]. Our study found that PCV2 infection induces the upregulation of IFN-β expression. This finding is consistent with the phenomenon in studies that SOCS3 inhibits interferon signaling such as in hepatitis C virus (HCV) and PRRSV, indicating that PCV2 may achieve immune escape through a similar mechanism.

Second, the regulation of inflammatory responses by SOCS3 also affects the infection process of PCV2. It has been found that SOCS3 overexpression can inhibit TNF-α mediated IκB-α degradation, block NF-κB nuclear translocation, and thus, attenuate inflammatory responses [32]. In contrast, when SOCS3 is silenced, TNF-α expression levels rise, NF-κB signaling is activated, and viral replication is inhibited. These results are consistent with previous studies on PCV2 subclinical infection that, namely, SOCS3 creates an immunosuppressive microenvironment conducive to persistent virus infections by inhibiting IL-6 and TNF-α signaling pathways. It should be noted that the competitive inhibition of the IFN-β gene enhancer by NF-κB may partly explain the upregulation of IFN-β expression in SOCS3 overexpression in this study.

Of particular interest is that PCV2 exhibits a unique utilization strategy for the interferon response. It has been shown that the Rep promoter region of PCV2 contains the interferon stimulation response element (ISRE), which may be the structural basis of interferon promoting virus replication after virus infection [33]. In this study, SOCS3 overexpression increased viral replication and IFN-β expression simultaneously, suggesting that PCV2 may inhibit interferon signaling pathway through SOCS3 while using ISREs to gain replication advantages, forming a delicate immune regulatory balance.

In conclusion, this study reveals a dual regulatory mechanism of SOCS3 as a key node for PCV2 immune escape: on the one hand, it weakens the host antiviral defense by inhibiting the JAK-STAT signaling pathway; on the other hand, it suppresses inflammatory response by blocking NF-κB activation. These findings provide new insights into the molecular mechanisms of PCV2 subclinical infection and the potential targets for optimizing existing prevention and control strategies. However, the specific interaction mechanism between SOCS3 and PCV2 protein remains to be further clarified.

## 5. Conclusions

This study unveils the critical regulatory role of the cytokine signaling suppressor SOCS3 in PCV2 infection. The experimental evidence demonstrates that PCV2 induces the upregulation of SOCS3 expression in host cells, establishing an immune-tolerant microenvironment conducive to viral replication. Specifically, SOCS3 acts as a host cofactor to directly enhance viral genome replication. Concurrently, SOCS3 remodels cytokine dynamics by suppressing IFN-β-mediated antiviral innate immunity and downregulating TNF-α-driven proinflammatory responses, thereby compromising host defense mechanisms against viral clearance. This dual regulatory mechanism provides critical insights into PCV2 immune evasion strategies and the molecular basis of subclinical persistent infection. Furthermore, it highlights the central role of SOCS3 as a key node in virus–host interactions during immune-suppressive microenvironment formation. The findings not only advance the theoretical understanding of PCV2 pathogenesis but also establish a molecular foundation for developing novel host-targeted therapeutic strategies. These discoveries hold potential implications for controlling PCV2-associated diseases and mitigating the economic losses in swine production.

## Figures and Tables

**Figure 1 viruses-17-01081-f001:**
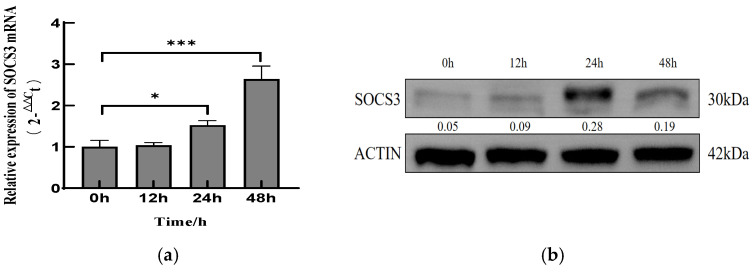
PCV2 infection upregulated the mRNA and protein levels of SOCS3 in PK-15 cells. (**a**) PK-15 cells were inoculated with PCV2 at a MOI of 1.0 or mock-infected for 0, 12, 24, and 48 h and analyzed by qRT-PCR measurements of the SOCS3 mRNA. (**b**) Immunoblotting with anti-SOCS3 and anti-β-actin antibodies to measure the SOCS3 protein at 0, 12, 24, and 48 hpi, respectively. Data are presented as mean ± SD (*n* = 3). * *p* < 0.05, *** *p* < 0.001 vs. control.

**Figure 2 viruses-17-01081-f002:**
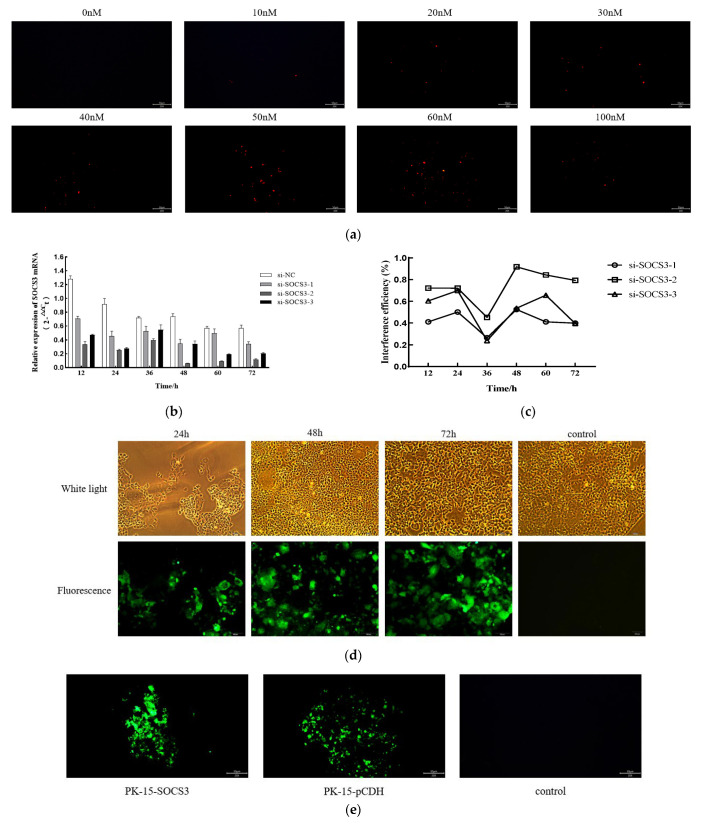
SOCS3 expression facilitates PCV2 replication. (**a**) Fluorescence imaging of PK-15 cells transfected with varying concentrations of Cy3-labeled siRNA. (**b**) SOCS3 mRNA expression levels and (**c**) corresponding knockdown efficiency at different time points post-siRNA transfection. (**d**) Fluorescence visualization of HEK-293T cells at distinct time intervals following lentiviral vector transduction. (**e**) Fluorescence patterns of SOCS3-stably transfected cells and blank controls after 10 consecutive passages. (**f**) Relative SOCS3 mRNA expression levels in stably transfected cells. (**g**) Cell viability analysis of SOCS3-silenced and overexpressing cell lines. (**h**) Protein expression profiles in SOCS3-knockdown and overexpression models. (**i**) Absolute quantitative PCR analysis of PCV2 viral copy numbers in wild-type PK-15, si-NC, and si-SOCS3-2 cells infected with PCV2 (MOI = 1) at indicated time points. (**j**) PCV2 viral load quantification in wild-type PK-15, PK-15-pCDH, and PK-15-SOCS3 cells post-infection. (**k**) Indirect immunofluorescence detection of PCV2 infection status across experimental groups. Data are presented as mean ± SD (*n* = 3). *** *p* < 0.001 vs. control.

**Figure 3 viruses-17-01081-f003:**
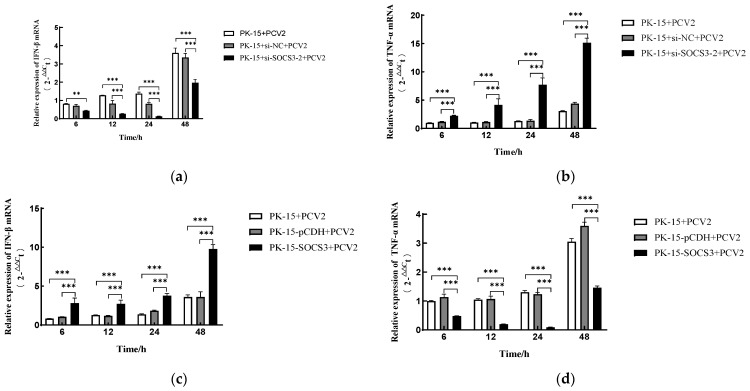
Effects of SOCS3 on PCV2-induced IFN-β and TNF-α mRNA expressions. (**a**) SOCS3 silencing suppresses IFN-β production in PCV2-infected host cells while (**b**) enhancing TNF-α expression. Relative quantitative PCR analysis of IFN-β and TNF-α mRNA levels in wild-type PK-15, si-NC, and si-SOCS3-2 cells infected with PCV2 (MOI = 1) at indicated time points. (**c**) SOCS3 overexpression promotes IFN-β production in PCV2-infected host cells, whereas (**d**) it inhibits TNF-α expression. Relative quantitative PCR analysis of IFN-β and TNF-α mRNA levels in wild-type PK-15, PK-15-pCDH, and PK-15-SOCS3 cells post-PCV2 infection (MOI = 1). Data are presented as mean ± SD (*n* = 3). ** *p* < 0.01, *** *p* < 0.001 vs. control.

**Table 1 viruses-17-01081-t001:** Primers used in this study.

Gene Product	Sense Primer (5′ to 3′)	Antisense Primer (5′ to 3′)
PCV2	ACCGTTACCGCTGGAGAAGGAAAAA	TGGTTACACGGATATTGTAGTCCTG
SOCS3	TCTAGAATGGTCACCCACAGCAAGTT	GCGGCCGCTTAAAGTGGGGCATCGTACT
qSOCS3	CGAGGCGAACCTGCTGCTT	ATCTTCAACACCCGCCTCT
qPCV2	GATGCGCAGGTTCTTGGTC	CAGGGCCAGAATTCAACCTT
β-actin	CTGTCCCTGTATGCCTCTG	ATGTCACGCACGATTTCC
qIFN-β	GCTAACAAGTGCATCCTCCAAA	AGCACATCATAGCTCATGGAAAGA
qTNF-α	CCTACTGCACTTCGAGGTTATC	GCATACCCACTCTGCCATT

Restriction enzyme sequences are underlined.

**Table 2 viruses-17-01081-t002:** The siRNA sequences.

Name	Sequence
si-SOCS3-1	GCUUCUCGCUGCAGAGUGAtt
	UCACUCUGCAGCGAGAAGCtt
si-SOCS3-2	GAAGAGCCUAUUACAUCUtt
	UAGAUGUAAUAGGCUCUUCtt
si-SOCS3-3	CCUGGACUCCUAUGAGAAAtt
	UUUCUCAUAGGAGUCCAGGtt

## Data Availability

All data generated for this study are included in the article.

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
