# Peer review of "PCV2 Infection Upregulates SOCS3 Expression to Facilitate Viral Replication in PK-15 Cells"

_viruses, 2025, doi:10.3390/v17081081_

Round 1
Reviewer 1 Report
Comments and Suggestions for Authors
- Line 38, the first letter of "porcine" should be capitalized.
- Line 86, the "2" in CO2 should be changed to an index.
- Line 100, "E. coli" should be changed to italics.
- Line 182, "/well" should be "cells/well". In Figure 2 (a), "nm" should be "nM". In the PDF document, figures (c), (g), (i), and (j) in Figure 2 are not shown.
- Line 309, there should be one character space between "IFN-β" and "mRNA".
- Line 324 should be Figure 3, not Figure 2.
- Line 352, the punctuation mark in "This" should be a period.
- Line 354, "porcine reproductive and respiratory syndrome virus" can be represented directly by "PRRSV", but there should be an abbreviation in line 341.
- Line 369, "[33][34]" should be "[33-34]".
- The format of references is not uniform. Some documents lack page numbers, and some Chinese documents do not mark in Chinese.
- Adding a graph to show the results will help readers better understand the results.

Author Response
请参阅附件。

Reviewer 2 Report
Comments and Suggestions for Authors
Major revisions:
Discuss whether the findings apply only to the PCV2d genotype or also to others like PCV2a and PCV2b, especially given recent studies showing PCV2d predominance and new mutations that could affect vaccine efficacy.
Minor Revisions
Define all abbreviations, such as DH5α, HEK-293T, RIPA, TBST, and PVDF, at their first use to ensure clarity for all readers.
Check that terms like "PCV2" and "PCV-2" are used consistently throughout the paper.
Proofread for spelling and grammar errors, especially in key sections like the abstract and discussion.
Update references with recent PCV2 research, including a 2025 study on genetic variation and a 2024 study on a PCV2d vaccine, to keep the paper current.
Reviewer 3 Report
Comments and Suggestions for Authors
The authors in the manuscript entitled "PCV2 Infection Upregulates SOCS3 Expression to Facilitate Viral Replication in PK-15” describes about the role of SOCS3 ISG expression in PCRV2 infection by overexpression and knockdown studies. The authors have also studies the expression of IFN-β and TNF-α expression. The reviewer has few suggestions for the improvement of the study and the manuscript, the authors need to address in their revised manuscript.
- The authors must clearly state the significance of their study in context of previous publications and what is the difference in the present study and what they want to achieve. The authors have not cited previous references in the introduction section.
- The authors have mentioned that “Notably, PCV2 infection upregulates SOCS3 expression in both peripheral blood lymphocytes and PK-15 cells, yet the mechanistic basis and virological significance of this modulation remain uncharacterized.”
The authors have not shown mechanistic insights as only overexpression and knockdown studies do not show any mechanistic insights. The authors must show direct interactions by interactomics (co-immunoprecipitation and western blot) and rescue the phenotype of the virus in knockout cells.
- The interferon response should be carried out with a functional interferon assay (not only qPCR of mRNA transcripts) like a plaque reduction assay or through model virus (vesicular stomatitis virus reporter) reporter system [The authors can refer- PLoS ONE 6(10): e25858. doi:10.1371/journal.pone.0025858)
Author Response
请参阅附件。

Round 2
Reviewer 3 Report
Comments and Suggestions for Authors I have checked the authors replies to my comments and I am now satisfied with their replies. The manuscript may be accepted for publications. Kind regards,